# The Influence of Media Coverage on the Negative Perception of Migrants in Chile

**DOI:** 10.3390/ijerph19138219

**Published:** 2022-07-05

**Authors:** Andrés Scherman, Nicolle Etchegaray, Isabel Pavez, Daniela Grassau

**Affiliations:** 1Department of Communication, Universitat Pompeu Fabra, Roc Boronat 138, 08002 Barcelona, Spain; 2School of Communications and Journalism, Universidad Adolfo Ibáñez, Diagonal Las Torres 2640, Peñalolén, Santiago 7941169, Chile; 3Millennium Nucleus Center for the Study of Politics, Public Opinion and Media in Chile (MEPOP, NCS2021_063), Vergara 323, Santiago 8370139, Chile; nicole.etchegaray@udp.cl (N.E.); mipavez@uandes.cl (I.P.); dgrassau@uc.cl (D.G.); 4School of Communication, Universidad Diego Portales, Vergara 240, Santiago 8370067, Chile; 5School of Communication, Universidad de los Andes, Mons. Álvaro del Portillo 12455, Las Condes, Santiago 7550000, Chile; 6School of Communications, Pontificia Universidad Católica de Chile, Avda. Libertador Bernardo O’Higgins 340, Santiago 8331150, Chile

**Keywords:** migration, well-being, mass media, Chile, framing

## Abstract

How mass media frame information about migrants is vital to further their well-being and social inclusion in the host society. However, it can also encourage forms of discrimination and xenophobia. In Chile, migration is a theme of great relevance due to a substantial increase in the flow of migrants over the last ten years, as well as the acts of extreme violence toward this group. Thus, this research analyzes how mass media coverage of migrants influences Chileans’ perception of migrants. Adopting an experimental design, we implemented a large-scale, representative online survey with citizens over 18 years of age (N = 2227) and an experiment with combined access to positive and negative news about migrants in different news outlets. The regression analysis showed differences between citizens regarding the type of daily relationships they are willing to maintain with the migrant population. Furthermore, exposure to news about migration was the primary variable that explains why people consider this phenomenon one of the main problems affecting the country, confirming the agenda-setting theory. The experiment also indicated a causal relationship between the exposure to negative content and negative judgment about this group and their presence in the country.

## 1. Introduction

The migratory context in Chile is of high interest. There has been an increase in international immigrants in the last decade [1,2,3], mainly from the rest of Latin America [4,5,6]. Migration is especially relevant in Chile because foreign nationals increased from 2% to 7.6% of the total population of the country between 2002 and 2021, mainly due to the arrival of Venezuelan, Colombian, and Haitian immigrants [7]. Recently, a sharp increase in illegal crossings in the northern part of the country has increased conflict and negative attitudes toward immigrants. There have been attacks by the Chilean population on immigrants who are in precarious situations and are living on public roads [8], which has a negative impact not only on foreigners’ lives but on social cohesion as well [9,10]. Thus, there is a complex scenario in part because migration is a recent cultural phenomenon where non-Chilean nationals are signified as outsiders regardless of their legal status or nationality [11,12]. Furthermore, it has fueled situations where migrants’ rights are neglected and they are not treated as equals by the Chilean society [13]. For the purposes of the paper, we operationalize the term migrant by referring to the definition upheld by the United Nations, which is “any person who is moving or has moved across an international border or within a State away from his/her habitual place of residence, regardless of (1) the person’s legal status; (2) whether the movement is voluntary or involuntary; (3) what the causes for the movement are; or (4) what the length of the stay is” [14]. Although this new generation of migrants has triggered public policies to advance their welfare and social integration [6,15], they tend to have failed to successfully integrate due to prejudice, vulnerability, exclusion [16,17], and deliberate acts of violence towards this group [18]. The evidence also reveals media messages linked to negative stereotypes of migrants [19,20,21]. Consequently, evidence of migrants’ experiences points to a sense of exclusion and alienation [22], depression, anxiety, and stress for those who legitimately seek to be part of Chilean society [20].

From a cultural perspective, the mass media as a social institution has a role in community formation, reaffirming ties, and sharing common values [23,24,25]. These practices could increase migrants’ sense of belonging [26,27,28], providing a common language and references, as well as patterns, social orientations, and beliefs that allow coexistence in society [25,29,30,31]. However, it can also encourage the contrary, such as discrimination, xenophobia, and exclusion. The first set of practices contribute to facilitating social integration, whereas the latter set of practices undermine the process of social integration [32]. Social integration is defined as a process whereby newcomers have a degree of connection with the host society that allows them to claim that they belong in it [33]. In this regard, the mass media has a crucial role to play as they influence the beliefs and values of the population (see, e.g., [34,35]). Therefore, the main goal of this study is to explore the effect of the mass media on Chileans’ perceptions of migrants, in a context where migration is still a relatively recent phenomenon in the country. We hypothesize that the mass media has a strong influence on migrants’ social integration.

### 1.1. Framing and Agenda Setting: The Influence of Mass Media on Perceptions and Attitudes

Since the 1980s, the concept of framing has been one of the most used constructs to study the sociology of communications [36]. Consensus in the literature has shown that frames in informative stories can refer to (1) the particular way in which a story is composed to optimize the accessibility of the audience at a given time/limited space [34]; (2) the selection and emphasis of certain elements and the marginalization and minimization of others [37,38,39]; (3) a central organizing idea that defines the problem and the context [40]; and (4) deep value and ideological frameworks [41] that are inserted in a network of discourses with which they dialogue [42]. One of the most cited definitions of the concept of framing is the one proposed by Entman, who states that “framing essentially involves selection and salience. To frame is to select some aspects of a perceived reality and make them more salient in a communicating text, in such a way as to promote a particular problem definition, causal interpretation, moral evaluation, and/or treatment recommendation for the item described” [38].

There is a growing consensus that the mass media portrayal of migrants disseminates and reinforces prejudices and negative attitudes towards this group (see, e.g., [43,44]). This is because of the effect of a range of negative framings on audience perceptions, knowledge, attitudes, or even emotions [36,37,45]. Early findings indicated that audiences could not process the realities in all their complexity [46,47]; thus, they attempt to infer causal relationships from sensory information [36]. These frames of interpretation incorporate rules for processing reality and provide guidelines for future decoding and behavior [46]. They also provide a structure of interpretation that, once learned, reduces the complexity of the comprehension processes and can be applied in the elaboration of messages shared in social interaction [36].

On the one hand, the effects of frames on audiences would be conditioned, among other things, by their prominence or salience (position and repetition), by their cultural resonance (adjustment with existing schemes) [38], their accessibility in memory, and their applicability in the processing of immediate knowledge [48]. On the other hand, the effect of frames can also be conditioned by personal characteristics such as experience with the topic covered, levels of exposure to the media, and previous attitudes about the topic. This is demonstrated, for example, by the work of Druckman and Chong [49], which shows that the repetition of certain frames would have a greater impact on people with less knowledge and who are also more attentive to peripheral signals, while individuals with more knowledge would be more likely to engage in systematic information processing by comparing the relative strength of alternative frames in competitive situations.

Despite more studies on the effects of frames, several questions remain. Firstly, there is debate regarding the independence between the notion of framing and that of agenda setting [50]. Agenda setting, in very general terms, predicts a high degree of convergence between the agendas of the different news media and the public agenda, specifically regarding the importance of specific issues, political figures, and other objects of attention and the presence of attributes that are assigned to said objects [51]. In other words, from the perspective of the agenda-setting theory [50,52], media help public opinion to learn about an issue and determine the importance that it will assign to it: the elements frequently highlighted by the media become the elements on which public debate focus [51]. While the media agenda is measured based on the salience that they give to certain issues in their news, the public agenda has been defined as those issues of public interest that people evaluate as the most important and is usually operationalized based on the following question: “What do you think is the most important issue facing our country today?” [51,53]. Although the view of dependency between both theories (framing and agenda setting) is strongly resisted by the authors dedicated to the subject, who propose the existence of an autonomous theoretical conceptualization [41,54,55], this paper proposes that both perspectives are complementary and can help explain the role of the media in the perception of migrants in Chile.

### 1.2. Negative Media Portrayal of Migrants

The frames given by mass media news on migration have been extensively studied in developed countries through content analysis, confirming the tendency to link migration with delinquency, crime, terrorism, and other social problems, while the information on its positive contributions is delivered to a lesser extent [43,56,57,58,59,60,61]. Conflict is a prevalent frame, comprising news where migrants are typically portrayed as victims or criminals [22,60,61,62], as a challenge for border security [63], or as a major economic and social cost for the state [55,59,64,65]. Likewise, studies examining news on Hispanic immigration in the United States reveal that foreigners are shown as lazy, ambitious, and/or as a threat to national stability and Americans’ job security (see, e.g., [53,62,66,67,68]). Similarly, Lakoff and Ferguson [54] analyze the framing of issues related to migration in the speech of former President George Bush as it was presented in the media. Among its conclusions, the focus on migration was as a problem that requires a solution, and as a detriment to the nation-state by utilizing frameworks such as globalization, humanitarian, civil rights, cheap labor, or economic refugees. The authors also highlight the superficial frameworks applied: equating migrants with illegality (for example, undocumented aliens), where “others” invade the country and threaten national security.

Similarly, in Chile, news framing has shown that migrants from Latin American countries tend to be linked to violence, crime, or negative portrayals, in contrast to those from Europe who are associated with positive news such as progress [19]. Scherman and Etchegaray [21] offer a deeper insight. They analyzed different media platforms covering print press, television, radio, and the Internet, finding that not all Latino migrants have the same press treatment and a new trend where migration is linked to conflict and political discussion. Latter research on mass media effects shows that the frames given to immigration have a significant influence on citizens’ attitudes, opinions, and emotions [45,49,67,69]. Igartua and colleagues [45,49] consider that the effects of frames may be governed by what they call heuristic processing, assuming that when media consumers are not mainly motivated to process information, the presence of peripheral cues in the journalistic message may condition cognitive responses. In the case of migration coverage, they confirm that group cues, that is, references to the national, ethnic, or geographic origin of migrants, moderate the effects of framing. In their study, the reference to Moroccan migrants increased the perception of migration as a problem and a negative attitude towards immigration [49]. The opposite was confirmed when the news referred to Latin American migrants.

The current study builds on the previous research on the effects of frames on citizens’ perceptions. We examine the following three questions:

RQ1: Are there differences between citizens regarding the type of relationships they are willing to maintain on a daily basis with the migrant population (living in a neighborhood with many immigrants, renting out their house or apartment to immigrants, working with immigrants, accepting their child marrying an immigrant, having their child in a school with many immigrant children, among other situations)?

RQ2: Is there a relationship between exposure to news about the situation of immigrants and the presence of this group of people in Chile as one of the issues that most affect Chileans (agenda-setting effect)?

RQ3: Is there a relationship between exposure to news with negative frames about immigrants and having a negative judgment about this group and their presence in the country (framing effect)?

## 2. Materials and Methods

### 2.1. Sample

In this study, we used a methodology that combined a survey with a large sample size (N = 2227 cases) and an experimental design. An online survey was used to do an experiment where researchers manipulated the independent variables of interest—journalistic texts with positive and negative thematic framing about immigrants on different information platforms—which were built considering the results of an earlier content analysis stage in which 400 news reports were reviewed. This content analysis has already been validated and its results have been published [21]. In this way, this research incorporates “experimental realism”, which has been developed in other research with the goal that the experiments have greater external validity and can be generalized due to the combination of public opinion studies and the use of media content as experimental stimuli [68,69,70,71]

This type of experiment using surveys allowed us to approach our two following research questions: RQ2, the existence of an agenda-setting effect, and RQ3, the presence of a framing effect.

To conduct this experiment, we surveyed a total of 1800 people online (79.1%) and 477 in person (20.9%). The same experimental conditions and stimuli were used for both groups (news stories that dealt positively or negatively with migrants, coming from print media, websites, and radio; a control group that received no stimulus was also included); for more information, see Appendix A and Appendix B. The study included people living in all regions of the country.

The data were gathered between December 2020 and January 2021. This was a period with a low number of legal entries by foreigners, due to the pandemic, but during which there was a rise in illegal entries according to testimonies from border zone residents.

### 2.2. Experimental Design

Study participants were exposed to the experimental condition at the start of the questionnaire and were subsequently asked questions about the news story they read or heard to verify proper comprehension. People who did not correctly answer basic questions about the initial stimuli they received were not considered in the final analysis. The experiment considered seven conditions (see Table 1). In three of these, the respondents were exposed to a positive story (from radio, newspapers, or websites) while in another three they were exposed to negative stories from the same platforms. The seventh group received no stimulus and served as a control group. After exposing the respondents to their respective stimuli, they were asked different questions about foreigners’ situation in Chile, ranging from the importance of this matter in public affairs to their level of favorability or negativity towards immigrants. The stimuli used can be seen in Appendix A. Prior to the experiment, we pre-tested that the positive and negative stimuli about immigrants were perceived in that way by people who were part of the survey target audience.

As part of the empirical analysis, we first carried out a cluster analysis using a battery of questions in which the interviewees were asked if they would accept or not maintain seven different types of relationships with immigrants. This analysis allowed a first descriptive approach with the aim of answering RQ1 and showing the types of relationships that respondents are willing to have with migrants.

### 2.3. Dependent and Independent Variables

In the regression analyses we used two dependent variables:

*Agenda setting and migration*. To measure the agenda-setting effect, we applied a variable where respondents were asked, “What is the principal problem currently affecting Chileans? What is the second biggest problem?” With these two answers, we created a dichotomous variable where value 1 was assigned to all respondents who mentioned immigration as the principal problem in their first or second choice, and value 0 was assigned to the rest of the sample (1 = 19.5%; 2 = 80.5%).

*Frames and migration.* The framing effect is measured by observing whether the impact of news with negative framing about migrants generates negative perceptions about this group. To construct this variable, we considered 10 ordinal variables asking about the level of agreement, on a scale of 1 to 5, with various affirmations about immigrants. These 10 affirmations were recodified and converted into binary variables (answers considering that immigrants’ presence is not generating difficulties in Chile were coded with 1, and the other responses were coded with 0 in each variable; a counter was subsequently built to add up the responses from each respondent with a 1 value, that is, those which considered that migrants were not generating difficulties). Theoretically, the counter had a minimum value of 0, equivalent to having no positive opinion about migrants and a maximum value of 10, which would mean having positive opinions about migrants in each dimension surveyed (M = 3.3; SD = 2.9). This counter was subsequently used as a dependent variable in a Poisson regression presented in the Results section. Appendix B presents the details of the 10 variables used to build this dependent variable and its descriptive statistics.

The two principal independent variables related to the experiment performed were as follows:

*Exposure to information*. We used this variable to measure the existence of an agenda-setting effect, and separated the sample between people exposed to information about immigrants (positive and negative) and the control group which received no information about the topic (exposed to information = 81.8%; not exposed to information = 18.2%).

*Exposure to different frames*. This variable was used to measure the framing effect and was separated into three categories: exposed to positive framing about migrants (36.8%), exposed to negative framing about migrants (45%), and not exposed to any framing about migrants (18.2%).

Considering prior research, we decided to incorporate four additional control variables:

*Evaluation of the economic situation*. To measure perceptions of the economic situation, we asked, “In your judgment, currently, is the Chilean economy progressing, stagnant or declining?

*Ideological position*. This variable position was measured on a scale of 1 to 10, where 1 was the left and 10 was the right.

*Degree of closeness with immigrants living in the country*. To establish closeness level with immigrants in the country, we asked the respondents if they had any current contact with foreigners in five different situations, and built a counter with a minimum value of 0 and a maximum of 5. We asked whether the respondents had current contact with immigrants in the following five situations: (a) within their families; (b) among their friends; (c) in their workplaces; (d) in their places of study; and (e) in their neighborhoods.

*Fear of crime*. To measure the different levels of fear of crime we used the following question: “Currently, how afraid are you of being a victim of a crime?” The responses were on a Likert scale with 1 being “no afraid” to 4 being “very afraid”.

Finally, we added three sociodemographic variables to the models: *gender*, *age*, and *social class*.

## 3. Results

To answer our first research question and evaluate the differences between citizens regarding the type of everyday relations they would be willing to have with the migrant population, we conducted a cluster analysis based on a battery of questions evaluating respondents’ disposition to voluntarily relate with immigrants in seven different areas (1. living in the same neighborhood where many immigrants live; 2. renting a house or apartment to immigrants; 3. working or studying with immigrants; 4. having an immigrant be your boss at work; 5. letting your child bring immigrant friends home; 6. having a son or daughter marry an immigrant; and 7. having your child in a school where there are many children of immigrants). Response options were 1 = I would accept; 2 = I would try to avoid it; and 3 = I would reject it.

Using the two-stage method of cluster identification, we first conducted an inductive inspection of the segmentation using the two-step method. This allowed us to identify the number of segments to define, according to the criterion of an increase in the BIC index based on the possible models’ fit. As a result, we defined the modeling of four clusters, whose description is detailed in the Results section. The segments’ creation was developed based on the K-medians clustering method.

The clusters suggest an affirmative answer to RQ1 and indicate that attitudes towards migrants in Chile are segmented into four groups, with different levels of openness towards interaction with foreigners depending on the situations described in the battery of seven variables detailed in the previous section (see Table 2).

The first cluster (N = 1306) includes the largest number of individuals and is characterized by accepting relations with immigrants in all the social spaces mentioned. The second cluster (N = 572) is only different from the first in two variables: members of this group would try to avoid living in a neighborhood with a high number of immigrants and would also avoid renting out a home to any members of this group. The third cluster (N = 279) is characterized by avoiding contact with immigrants in all situations mentioned, as well as rejecting the idea of renting a home to foreigners. The final cluster (N = 96) concentrates the individuals with the least inclination to relate with immigrants, as they reject every form of interaction with foreigners that was polled.

To complement the clusters’ descriptive analysis, we conducted bivariate analyses (ANOVA and Spearman’s rho and Somers’ d tests) to establish whether differences existed between the clusters according to the different control variables included in the regression analysis (see Table 3 and Table 4).

This analysis shows that in all the statistical tests, the relation with immigrants was linearly ordered (from lowest to highest) for the majority of the variables analyzed. The average for the political positioning variables (1 = left; 10 = right) was significantly different between groups (F = 50.535; *p* = 0.000), with cluster 1, on average, being more towards the left (M = 44.60; SD = 2.18), while the highest average was found among individuals in cluster 4 (M = 6.59; SD = 2.10). A similar pattern is observed with age: people in cluster 1 had an average age of 44.6 years (SD = 15.3), while the median in cluster 4 was 50.4 years (SD = 16.6).

The socioeconomic level variable followed the pattern indicated. Cluster 1 had the largest proportion of people with greater purchasing power (upper class = 43%; lower class = 19.4%) while cluster 4 had the largest proportion of people with fewer resources (upper class = 21.6%; lower class = 40.9%), with an equally significant association between variables (Spearman’s rho = −0.073, *p* = 0.000; Somers’ d = −0.076, *p* = 0.000).

The results of Table 5 allow us to answer the second research question of this study. Exposure to news related to immigrants influences whether individuals view immigration as one of the most relevant issues in the country. Exposure to both positive and negative framings about migrants increases the chance of people considering immigration to be one of the two most important issues for the country to handle (The dependent variable for this analysis is categorical and has only two values: 1 (immigration is considered one of the most relevant issues on the country’s public agenda) and 0 (immigration is not considered one of the most relevant issues on the public agenda). The negative and significant coefficients express a decrease in the chance of the presence of the dependent variable when the independent variable is modified; in the same way, negative and significant coefficients express an increase in the chance of the presence of the dependent variable in the same condition).

Additionally, the results show that there is a positive relationship between having more personally intense or close relations with immigrants and whether this subject should have a more relevant spot on the media agenda. Similarly, those who were more afraid of becoming crime victims, those who consider that the economy is stagnant or declining, and those who have right-wing political positions were more likely to view migration as a highly relevant media agenda topic. By contrast, being male, being in a high socioeconomic level, and being 65 or older decreases the probability of someone considering that immigration is a priority subject for the public agenda.

Descriptive statistics of the independent variables are: *evaluation of the economic situation* (progressing = 12.5%, stagnant = 58.2%, declining = 29.2%), *ideological position* (M = 5.18, SD = 2.3), *degree of closeness with immigrants living in the country* (M = 1.31, SD = 1.18), *fear of crime* (no afraid = 4.1%, low fear = 28.7%, somewhat of afraid = 40%, and very afraid = 27.3%), *gender* (women = 52%), *age* (M = 46.4 years, SD = 15.6), and *social class* (lower class = 22.2%, middle class = 39.2%; upper class = 38.7%).

Our third and final research question explored if there is a relationship between exposure to news with negative frames about immigrants and having a negative judgment about this group and their presence in the country (framing effect). The results in Table 6 answer this research question. Exposure to news with negative framing about immigrants is a reason to disagree with the idea that foreigners’ presence does not generate problems for the country. By contrast, exposure to positive framing about immigrants has no effect on the mode of evaluating the consequences of the presence of this group.

Other variables related to disagreeing with the idea that immigrants’ presence does not generate problems include being older (50 years or older), having right-wing political positions, considering that the economy is stagnant or declining, and being afraid of becoming a crime victim. By contrast, the variables associated with agreeing that foreigners do not cause problems included belonging to medium or high-income groups and having frequent personal relations with immigrants in different aspects of social life.

## 4. Discussion

### 4.1. Key Findings and Implications

This study provides relevant empirical support for two important theoretical perspectives on mass media effects on public opinion, agenda setting and framing, applied to a central phenomenon that affects many countries around the globe and millions of human beings: the growing flows of international migration. The results demonstrate the influence of news content on the attitudes and opinions of the national audience. These conclusions are the result of a rarely implemented methodological design, which consists of using an experimental strategy in the context of a survey in a country that has experienced an accelerated and growing influx of immigrants from various parts of Latin America in a short period of time. This is important because, in the field of news framing research, past studies have used experiments similar to a laboratory condition [31,52,60,65,67,69,72], obtaining data with valuable internal validity, but with little external validity and no possibility of representing the general population or identifying the differences among the segments studied.

One of the central purposes of this study was to utilize a more robust methodological approach, implementing an explanatory analysis strategy over a representative sample of the general population in Chile. Consistent with previous studies, the results show that media content can influence the audiences in two different ways: the importance given to a subject matter, the agenda-setting effect [32,36], and the way they think about this subject, or the framing effect [34,40]. In each way, the representativeness of the data allows other relevant determinants of people’s opinions to be identified. Along with exposure to news about migrants, other determining factors for considering whether immigration is one of the main issues affecting the country (agenda-setting effect) are: (a) being part of the youngest segments, (b) belonging to lower-income groups, (c) expressing conservative political positions, (d) feeling very afraid of being a victim of a crime, and (e) maintaining frequent contact with foreigners in different aspects of daily life.

The study also contributes to understanding the role of mass media on citizens’ attitudes on immigrants. Results are consistent with previous literature [31,64,65,67,72], showing that people exposed to news with a negative frame of migration express significantly more intolerant attitudes toward immigrants than those exposed to positive or neutral frames. It is interesting to observe that positive frames do not show an influence on attitudes towards immigrants, something that, although contrary to what the theory would predict, is consistent with previous studies such as that of Kim and colleagues [73]. These researchers reported that news content that shows immigrants as a real or symbolic threat generates negative emotions and attitudes, but non-threatening messages also show a small negative effect on their emotions. In the same way, coefficients of both frames in our analysis were negative, but the relation of the positive frame did not reach statistical significance.

The representativeness of the sample allows us to establish that attitudes toward immigrants are not homogeneous. Attitudes range from a majority group that is highly open to immigration (83%) to a minority group that shows a strong rejection to interacting with foreigners in daily life and to sharing different spaces with them (17%). The composition of these groups is associated with variables such as age, SES, political position, and fear of being a victim of a crime. The most reactive groups towards immigrants are, on average, those with a lower socioeconomic status (less educated and with fewer economic resources), which is consistent with previous literature on xenophobia determinants. These studies highlight economic competition as a strong trigger of negative attitudes toward immigrants [58,59].

On the other hand, the association between attitudes towards migration and the political position of individuals has also been confirmed by numerous studies [60,74,75]. Greater levels of fear of crime being associated with anti-immigrant views is consistent with content analysis research that has shown the tendency of news to link immigrants with conflicting events and concepts such as illegality and criminality in Chile [61], as in many developed countries (see, e.g., [61,62,63]).

### 4.2. Limitations

Despite its valuable findings, this study has a number of limitations. First, the field work was carried out during the period of the COVID-19 pandemic. As a result, the massive influx of immigrants into the country was halted for a few months and it caused a decrease in migratory pressure in some areas of the country. Nevertheless, in general the number of foreigners living in Chile did not decrease during the pandemic (National Statistics Institute—INE, 2021). Although this situation did not affect the results of the experiment, the problems associated with field work during the pandemic meant we needed to use an online survey rather than a face-to-face approach as we had planned. This may have caused two problems: (a) less sample representativeness in terms of the characteristics of the country’s population and (b) a self-selection effect which increased the interest in survey participation among people who either reject the presence of immigrants or support their presence in Chile.

## 5. Conclusions

This study provides evidence regarding the effect of the media on the perception of the Chilean population of migrants in the context of a country that has increased the number of immigrants in a short period of time. The negative representation of migrants shown by the media can lead to acts of xenophobia, which can further harm the precarious situation of many foreigners who have come to Chile from other Latin American countries. Such results may influence a reflection among actors in the media about the way they report on immigrants in Chile and among other stakeholders linked to the issue, such as immigration authorities and civil society organizations.

## Figures and Tables

**Table 1 ijerph-19-08219-t001:** Experimental conditions (platforms, framing) and number of cases.

Condition	Platform (a)	Predominant Framing (b)	Treatment	N
1	Print media (1)	Positive (1)	a_1_b_1_	300
2	Print media (1)	Negative (2)	a_1_b_2_	348
3	Web (2)	Positive (1)	a_2_b_1_	335
4	Web (2)	Negative (2)	a_2_b_2_	313
5	Radio (3)	Positive (1)	a_3_b_1_	377
6	Radio (3)	Negative (2)	a_2_b_2_	265
7 (Control)	Neither (0)	None (0)	a_0_b_0_	338

**Table 2 ijerph-19-08219-t002:** Cluster analysis.

In Each Case, Would You Accept This Type of Relation (1), Try to Avoid It (2), or Reject It (3)?	Cluster			
	1	2	3	4
P15.1 Live in a neighborhood with many immigrants	2	1	2	3
P15.2 Rent out a house or apartment you owned to immigrants	3	1	2	3
P15.3 Work/study with immigrants	2	1	1	3
P15.4 Have an immigrant be your boss at work	2	1	1	3
P15.5 Have your child bring immigrant friends home	2	1	1	3
P15.6 Have your child marry an immigrant	2	1	1	3
P15.7 Have your child in a school with many immigrant children	2	1	1	3

**Table 3 ijerph-19-08219-t003:** ANOVA of association between clusters and numeric variables.

	Age (Years)		Political Position (Left = 1; Right = 10)	
	Median	SD	Median	SD
Cluster 1	44.60	15.353	4.70	2.185
Cluster 2	47.24	15.171	5.72	2.055
Cluster 3	50.61	15.408	6.06	2.651
Cluster 4	50.41	16.600	6.59	2.108
Total	46.23	15.513	5.19	2.298
F statistic	15.295		50.535	
Sig. (bilateral)	*p* = 0.000		*p* = 0.000	
N	2222		1984	

**Table 4 ijerph-19-08219-t004:** Association between clusters and SES.

	Low	Middle	Upper	Total
Cluster 1	19.4%	37.6%	43.0%	100%
Cluster 2	24.0%	42.4%	33.6%	100%
Cluster 3	22.0%	41.2%	36.9%	100%
Cluster 4	40.9%	37.5%	21.6%	100%
Total	21.8%	39.2%	39.0%	100%
	Value	Sig. (bilateral)	N	
Spearman’s rho	−0.076	0.000		
Somers’ d (symmetric)	−0.073	0.000	2471	2160

**Table 5 ijerph-19-08219-t005:** Logistic regression of the determinants of agenda.

	B Unstandardized
	Coefficients
Men [Ref: Women]	−0.226
	(0.129)
Age: 35–49 [Ref: 18–34]	0.044
	(0.162)
Age: 50–64	−0.159
	(0.171)
Age: 65+	−0.718 ***
	(0.212)
SES Middle [Ref: Low]	−0.022
	(0.166)
SES Upper	−0.760 ***
	(0.182)
Political position [1 = Left; 10 = Right]	0.283 ***
	(0.030)
Economic situation: Stagnant [Ref: Making progress]	0.478 ***
	(0.183)
Economic situation: Declining	0.682 ***
	(0.218)
Low fear of crime [Ref: No afraid]	0.381
	(0.481)
Somewhat afraid of crime	0.824
	(0.468)
Very afraid of crime	1.519 ***
	(0.469)
Immigrant relations	0.095 **
	(0.045)
Exposure positive frame (Ref: Not exposed to news)	0.891 ***
	(0.218)
Exposure negative frame	0.799 ***
	(0.222)
Intercept	−4.911 ***
	(0.582)
Observations	1897
Log likelihood	−905.210
Akaike Inf. Crit.	1842.419

** *p* < 0.01; *** *p* < 0.001.

**Table 6 ijerph-19-08219-t006:** Poisson ^†^ regression of the determinants of agreement that the presence of foreigners is not causing problems.

	Agreement Presence of Foreigners
	(Unstandardized Coefficients)
Men [Ref: Women]	−0.016
	(0.0243)
Age: 35–49 [Ref: 18–34]	−0.035
	(0.0393)
Age: 50–64	−0.235 ***
	(0.0391)
Age: 65+	−0.085 **
	(0.0393)
SES Middle [Ref: Low]	0.151 ***
	(0.0394)
NSE Upper	0.409 ***
	(0.0386)
Political position [1 = Left; 10 = Right]	−0.139 ***
	(0.0058)
Economic situation: Stagnant (Ref: Making progress)	−0.161 ***
	(0.0294)
Economic situation: Declining	−0.316 ***
	(0.0429)
Low fear of crime [Ref: No afraid]	−0.118 *
	(0.0522)
Somewhat afraid of crime	−0.445 ***
	(0.0534)
High fear of crime	−0.732 ***
	(0.0585)
Immigrant relations	0.099 ***
	(0.0102)
Exposure positive frame (Ref: Not exposed to news)	−0.056
	(0350)
Exposure negative frame	−0.118 **
	(0358)
Intercept	2.166 ***
	(0.0800)
Observations	2133
Log likelihood	−4771.737
Akaike Inf. Crit.	9575.474

* *p* < 0.5; ** *p* < 0.01; *** *p* < 0.001, ^†^ The Poisson regression—and not a multiple regression model—was used due to the variable being a count data, composed of the sum of the questions in which the respondents agreed with some statements related to immigrants. The counter had a minimum value of 0 (equivalent to having no positive opinion about migrants) and a maximum value of 10 (M = 3.3; SD = 2.9). This type of variable does not have a normal distribution and does not meet the assumptions required by a multiple linear regression. For this reason, the literature recommends the use of a Poisson regression to analyze this type of data.

## Data Availability

The information from this study is in the hands of the researchers and can be made available to the journal, respecting the anonymity of the respondents, if its editors deem it necessary.

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
