# Peer review of "The Influence of Media Coverage on the Negative Perception of Migrants in Chile"

_ijerph, 2022, doi:10.3390/ijerph19138219_

Round 1
Reviewer 1 Report
The article is timely and relevant and such research must happen regularly to understand the public discussion. But I see some issues in this article, most of them even before the methodological design of the research.
The main issue I detect is with the definition of "immigrants". The introduction mentions too briefly that there are different social perceptions associated to the origin of the migrants; but after this point, the study seems to ignore completely these differences and keeps insisting on the word "immigrant" that is only defined in opposition to Chileans. I fear that this research approach might not only fault the questionnaire (do the respondents know what immigrants you are asking about? does it matter? How did you ensure that the questionnaire holds valid in front of this nuance?) Avoiding this issues might accidentally contribute to the problem rather than solving it; because it grows and expands on such a manichean distinction. The authors should try to find ways of dealing with this concept in the text; what scholarship is out there and what have other experiments done to solve these issues; and how has this been done in yours. It would be helpful then to clarify better how the authors have dealt with it in their experiment and research.
This confusion transpires then into the text and it leads to some strange statements such as the one in line 431 which seems speculative and does not appear to be connected to data (I suggest deleting it -or connecting it to data).
There are some translation issues, too; and all along the text some uses of English could benefit from another check. For instance paragraph starting in line 455 needs translation from Spanish.
Reviewer 2 Report
- The first two hypotheses need to be rewritten. as presently written, they make no sense. It was only by reading the results, that the test became clear.
- 65-69 The sentence is convoluted. rewrite.
- 277 Using a Chi square with such a large N is not very useful
- 321 Same as above
- Table 5 reveals some very low correlations. Comment on them
- You need to speak to the impact of "positive framing" in that you found "no effect". Why?
- No explanation as to why you used a Poisson regression model.
- Why not use a multiple regression model?
- The "Discussion is too long. shorten it.
- Show how the results comment on the two theoretical models--framing and agenda setting.
- 455-458 why in spanish?
- 19, 151, 162 all have different Ns. Explain why
Reviewer 3 Report
I thank the opportunity to review this article. Its reading has been satisfying because, although the research is based on a theoretical and methodological approach already widely used before. I found it satisfying to read because, although the research is carried out from a theoretical and methodological approach already widely used before, it is a work that can be considered novel due to the fact that there are not many studies in the Latin American context that address the effects of agenda setting and framing from experimental approaches. In this sense, I would like to congratulate the authors for their contribution. However, there are some aspects that I consider should be revised in the work to improve its quality.
-Firstly, and here lies the main problem I observe in the paper, this is a work where the famous expression "less is more" applies in a clear way. The article points out that two studies have been carried out: one by means of a survey to provide a typology among citizens to determine the acceptance of the relationship with immigrants and the other by means of an experiment to evaluate the effects of exposure to media content on the perception and attitude of the population. I consider that an attempt has been made to force the union of both studies in the same article, when they are two works with different approaches and scopes. In this sense, it is suggested to focus the current paper on the experimental study of media effects, which in the end is the one with theoretical support and on which the conclusions of the study are based.
-Once this adjustment has been made, I believe that the theoretical section of the paper should be improved. It is surprising that the abstract mentions the agenda-setting effect, but this theory is not described in the literature and, at the same time, the paper provides a broad background on framing theory, but this theory is not even mentioned in the abstract. I suggest re-elaborating the theoretical section, describing the theories, discussing their points of connection and presenting the existing background on media effect and immigration studies conducted from both theories. In fact, it is surprising that in the conclusions there are citations to previous studies that have not been described in the theoretical section; this should be resolved. Finally, it would be advisable to include a contextual section on immigration in Chile, something which appear spread throughout the literature section in the current version.
-I suggest clearly stating the aim of the study at the end of the introduction section, so that the article's reader has more clarity about what was expected to find with the study.
-The hypotheses of the study lack, at least in the current version, the theoretical and empirical support that would allow them to be proposed. What studies do the authors rely on to formulate them in the Chilean context? Perhaps it would be more interesting to work with research questions, as this study is one of the first, if not the first, on this subject in Chile.
-In the method section, I suggest highlighting one of the important contributions of the study. Although it is common to find experimental studies in the literature on framing effects, it is more difficult to find works that provide greater realism to the study of this process of media influence. This paper constitutes one of those cases, but this goes unnoticed because it is not highlighted in the writing of the document. I suggest reviewing studies such as the one by Muñiz and Echeverría (2020) in the journal "Profesional de la Información" and the literature cited by them, which describes the "experimental realism" that is increasingly demanded in this type of studies.
-As minor aspects, I suggest providing more detail about the variables used, so that each one occupies a different paragraph. I also suggest revising the use of the expression "foreigners" instead of immigrants in some places of the document, since both concepts refer to different ideas. Finally, the authors should revise the translation of one of the last paragraphs that appears in Spanish.
Round 2
Reviewer 3 Report
I thank the author(s) for having considered the recommendations made in my review. I consider that the article has improved significantly, but there are some minor aspects that should be corrected before its publication:
-The aim of research is explained at the end of the theoretical background, but it should really appear at the end of the first section dedicated to the introduction.
-The wording of the research questions should be revised, as they continue to be posed as hypotheses: "There are differences..." when it should be written "Are there differences..."
-The reference to hypotheses is maintained in the text when they have been modified to research questions. For example, when on page 4 the article states "following research questions: H2, the existence of an agenda setting effect; and H3, the presence of a framing effect.", instead of using RQ2 and RQ3.
-The wording of the article requires a general revision. There are drafting errors, such as when on page 9 the text states "reason to disagree t with the idea that" (that "t" is left out). Or when “d de Sommers test” is stated instead of "Somers' d test", or “rho Spearman” instead of "Spearman's rho".
